# Transcriptome Analysis and QTL Mapping Identify Candidate Genes and Regulatory Mechanisms Related to Low-Temperature Germination Ability in Maize

**DOI:** 10.3390/genes14101917

**Published:** 2023-10-08

**Authors:** Lei Du, Xin Peng, Hao Zhang, Wangsen Xin, Kejun Ma, Yongzhong Liu, Guangcan Hu

**Affiliations:** 1Institute of Upland Food Crops, YiChang Academy of Agricultural Science, Yichang 443011, China; 2College of Plant Science and Technology, Huazhong Agricultural University, Wuhan 430070, China; bzn@webmail.hzau.edu.cn (X.P.); maizezh@163.com (H.Z.); sen0607@126.com (W.X.); mazhenghang@163.com (K.M.); 3Hubei Hongshan Laboratory, Wuhan 430070, China; dulei@mail.hzau.edu.cn (L.D.); liuyongzhong1@mail.hzau.edu.cn (Y.L.)

**Keywords:** low-temperature germination ability, transcriptome analysis, WGCNA, quantitative trait locus, doubled haploid, candidate gene

## Abstract

Low-temperature germination ability (LTGA) is an important characteristic for spring sowing maize. However, few maize genes related to LTGA were confirmed, and the regulatory mechanism is less clear. Here, maize-inbred lines Ye478 and Q1 with different LTGA were used to perform transcriptome analysis at multiple low-temperature germination stages, and a co-expression network was constructed by weighted gene co-expression network analysis (WGCNA). Data analysis showed that 7964 up- and 5010 down-regulated differentially expressed genes (DEGs) of Ye478 were identified at low-temperature germination stages, while 6060 up- and 2653 down-regulated DEGs of Q1 were identified. Gene ontology (GO) enrichment analysis revealed that ribosome synthesis and hydrogen peroxide metabolism were enhanced and mRNA metabolism was weakened under low-temperature stress for Ye478, while hydrogen peroxide metabolism was enhanced and mRNA metabolism was weakened for Q1. DEGs pairwise comparisons between the two genotypes found that Ye478 performed more ribosome synthesis at low temperatures compared with Q1. WGCNA analysis based on 24 transcriptomes identified 16 co-expressed modules. Of these, the MEbrown module was highly correlated with Ye478 at low-temperature stages and catalase and superoxide dismutase activity, and the MEred, MEgreen, and MEblack modules were highly correlated with Ye478 across low-temperature stages, which revealed a significant association between LTGA and these modules. GO enrichment analysis showed the MEbrown and MEred modules mainly functioned in ribosome synthesis and cell cycle, respectively. In addition, we conducted quantitative trait loci (QTL) analysis based on a doubled haploid (DH) population constructed by Ye478 and Q1 and identified a major QTL explanting 20.6% of phenotype variance on chromosome 1. In this QTL interval, we found three, four, and three hub genes in the MEbrown, MEred, and MEgreen modules, of which two hub genes (*Zm00001d031951*, *Zm00001d031953*) related to glutathione metabolism and one hub gene (*Zm00001d031617*) related to oxidoreductase activity could be the candidate genes for LTGA. These biological functions and candidate genes will be helpful in understanding the regulatory mechanism of LTGA and the directional improvement of maize varieties for LTGA.

## 1. Introduction

Maize (*Zea mays* L.) originated in tropical and subtropical regions and is inherently sensitive to low-temperature stress [1]. For maize, 25 °C is the optimum temperature for seed germination, while temperatures below 15 °C may cause low-temperature stress [2]. Low-temperature stress usually leads to seed emergence delay, slow seedling growth, and ultimately grain yield decreasing in many maize-production regions [3]. In China, more than 30 million hectares of spring sowing maize may encounter low-temperature stress during seed germination [4]. Therefore, it is necessary to improve low-temperature germination ability (LTGA) for cultivated varieties using breeding and genomic approaches.

Low-temperature tolerance is a complex quantitative trait involving numerous genes and mechanisms. In maize, several genes have been confirmed to be associated with this trait, including Zm*DREB1/CBF*, *ZmRR1*, *ZmMPK8*, *bZIP68*, and *ZmICE1* [5,6,7,8]. Of these, the *ZmDREB1/CBF* transcription factor regulates the expression of the *DRE* (dehydration-responsive element) gene and results in higher low-temperature tolerance in seedlings [5]. *ZmRR1* encoding a mitogen-activated protein kinase (MAPK) positively regulates the expression of *ZmDREB1* to enhance low-temperature tolerance at the seedling stage, and *ZmMPK8* is the negative regulator of *ZmRR1* [6]. The *bZIP68* transcription factor could repress the expression of *ZmDREB1* and lead to poor low-temperature tolerance in seedlings [7]. *ZmICE1* transcription factor positively regulates low-temperature tolerance both in seedling and seed germination, and the mechanism is blocking mitochondrial reactive oxygen species (ROS) bursting or directly regulating the expression of *ZmDREB1s* [8]. In addition, several candidate genes associated with LTGA were identified, for example, *ZmbZIP113* and *ZmTSAH1* [9], *Zm00001d039219*, and *Zm00001d034319* [10]. These genes and mechanisms will be an important basis for genetic studies and directional variety improvement for LTGA in maize.

Except for that, these are other well-known regulatory mechanisms for low-temperature tolerance in plants. Plasma membrane functions, including fluidity and signal transduction, are widely considered to be correlated with plant low-temperature tolerance [11]. Membrane fluidity is affected by desaturated fatty acid content (DFC), and numerous genes have been found to modulate low-temperature responses by regulating DFC content [12,13,14]. Meanwhile, many signal channels in the plasma membrane are essential to transduce low-temperature sensing, and several genes involved in these channels have been confirmed to be associated with low-temperature tolerance [15,16,17]. In addition, low-temperature signal transmission intracellularly usually involves a variety of signal molecules and signal pathways. Plant hormones are well-known signal molecules for the regulation of low-temperature tolerance; for instance, increasing ABA biosynthesis could improve low-temperature tolerance [18], reducing free IAA content through overexpression of *OsGH3-2* could increase low-temperature tolerance [19], and cytokinin [20,21], ethylene [22,23], and other hormones have been found to be correlated with low-temperature tolerance. Meanwhile, intracellular signal transmission usually relies on the MAPK cascades, which contain three kinases and amplify the signal step by step via phosphorylation. To date, numerous genes involved in MAPK function have been found to be associated with low-temperature tolerance in Arabidopsis [24,25], maize [6], and rice [26,27]. Transcription factors (TF) receive upstream signals and promote the transcription of downstream genes. Several TFs are reported to be involved in low-temperature tolerance, for example, maize IF genes *ZmDREB1/CBF* and *bZIP68* [5,7], apple IF gene *NAC* [28], and rice IF genes *bZIP73* and *OsWRKY45–1* [29,30]. Except for these regulation mechanisms, there are a majority of other molecular mechanisms for low-temperature stress. Reactive oxygen (ROS) usually accumulates when plants encounter stress, such as low temperatures, and excessive ROS could lead to deleterious effects on plant cells, even cell death [31]. Therefore, the ROS-scavenging system was also an important mechanism for low-temperature tolerance. Of these, ROS-scavenging enzymes, including superoxide dismutase (SOD) and catalase (CAT), have been confirmed to increase low-temperature tolerance in plants [32,33]. Some chemicals, such as glutathione and ascorbic acid, could reduce ROS content to increase low-temperature tolerance in plants [34,35]. In addition, maintaining normal physiological function in cells under stress is important to withstand stress. Cell cycle is essential for the growth and development of plants, and some studies have found some genes regulate low-temperature tolerance by maintaining cell division in stress [36,37]. Similarly, ribosomes are essential for protein synthesis in living cells, and ribosomal synthesis and processing under low temperatures have been found to affect low-temperature tolerance, for example, ribosomal subunit synthesis-related genes *RPS5* and *CRASS* [38,39] and rRNA processing-related gene *STCH4/REIL2* [40]. All of these genes and regulatory mechanisms will be helpful in understanding the molecular basis and exploring excellent alleles for low-temperature tolerance.

Weighted gene co-expression network analysis (WGCNA) is usually used to research the gene expression patterns of multiple samples, which divide genes with similar expression patterns into the same modules. In addition, WGCNA can also analyze the relationship between modules and sample phenotypes and identify key regulatory modules and genes [41]. This method has been widely used to identify abiotic and biotic stress-responding genes in plants [41,42,43].

LTGA is an important characteristic for the early sowing of spring maize and high-altitude maize. The two inbred lines Ye478 and Q1 with different LTGAs were important maize germplasm sources for breeding in China. In this study, the objectives were to investigate the molecular basis of the LTGA difference between the two genotypes, explore the underling mechanism for LTGA, and identify key gene modules and candidate genes correlated with LTGA. To achieve these objectives, we conducted the transcriptome analysis for the two genotypes under low-temperature stress during seed germination, identified DEGs between the two genotypes and between the germination stages, and analyzed the biological function of these DEGs based on gene ontology (GO) enrichment analysis. In addition, we constructed a gene co-expression network based on WGCNA, identified key gene modules correlated with the samples and specific enzymatic activity, detected hub genes, and analyzed their underling biological function. In the end, we conducted QTL analysis based on the DH population constructed by Ye478 and Q1 and investigated candidate hub genes in the major QTL interval. These results will be used to understand the molecular basis of LTGA and further provide useful genes for directional breeding improvement.

## 2. Materials and Methods

### 2.1. Plant Materials

Two maize elite inbred lines were used to conduct transcriptome analysis in this study. The inbred line Ye478 was tolerant to low-temperature stress during seed germination, which has been widely applied over the past two decades. The inbred line Q1 was sensitive to low temperatures during seed germination and was bred from the cross of the commercial hybrid variety S7913 and the tropical landrace *Tuxpeno*. In addition, a DH population consisting of 123 DH lines constructed by Ye478 and Q1 as described previously [44] was used to perform QTL analysis for low-temperature germination rate.

### 2.2. Phenotypic Evaluation of Low-Temperature Tolerance during Germination

To ensure the quality and quantity of the seeds, the two inbred lines and the DH population were propagated in the spring of 2019 in Wuhan city. For the two inbred lines, the germination rate of the two inbred lines was evaluated with three replicates at low (12 °C) and optimum (25 °C) temperatures in the growth chamber, respectively. In brief, the seeds of each genotype were soaked in a 0.5% (*v*/*v*) hydrogen peroxide solution for 20 h at 25 °C. The seedling substrate was prepared using sterilized vermiculite with a 20% (*w*/*w*) distilled water content and put in a transparent germination box (19 × 13 × 12 cm). In each germination box, the thirty soaked seeds were uniformly placed on top of 2.0 cm of the seeding substrate and then covered with 1.0 cm. These prepared germination boxes were, respectively, located in the low and optimum temperature growth chambers, and 3000 lx light intensity was maintained through the whole germination stage. The germination rate for each germination box was investigated after emergence and repeatedly surveyed until the germination rate did not change. Seedlings are considered to emerge when they exceed 0.5 cm of the seeding bed. The germination rate for each germination box was the ratio of the seedling numbers to the seed numbers. For the DH population, the germination rates were evaluated with three replicates only at low temperatures in the growth chamber, and the germination rate was investigated 15 days after sowing. The phenotypic data of the two inbred lines and the DH population were organized in Excel 2010 and plotted and depicted using R software (version 4.3.1).

### 2.3. Measurement of Enzyme Activity

We measured the catalase (CAT) and superoxide dismutase (SOD) activity for the Ye478 and Q1 at four germination stages under low-temperature stress. The first sampling was taken after the seed soaking described in Section 2.2. The second, third, and fourth samplings were taken at the first, third, and fifth days after sowing, when the two genotypes were germinating at the low temperature conditions described in Section 2.2. At each sampling, the embryos of the seeds were dissected on ice and immediately frozen in a −80 °C refrigerator until enzyme activity was measured. Each sample consisted of two biological replicates, and each replicate contained ten embryos. The activities of CAT and SOD were measured with a commercial Boxbio kit (Beijing Boxbio Science Technology, Beijing, China).

### 2.4. RNA-Sequencing and Data Analysis

To explore the molecular mechanism of low-temperature tolerance during germination, the transcriptome analysis for Ye478 and Q1 was conducted at four stages under low-temperature stress. In brief, the seeds of the two genotypes were soaked in 0.5% (*v*/*v*) hydrogen peroxide solution for 20 h at 25 °C, and then the first sampling was conducted as the control, namely the 0d stage. The other soaked seeds of the two genotypes were, respectively, sown in transparent germination boxes described in Section 2.2, and then the germination boxes were put in a growth chamber at 12 °C and 3000 lx. Subsequently, the second, third, and fourth samplings were taken at the first, third, and fifth days after sowing and named as 1d, 3d, and 5d stages, respectively. At each sampling, the embryos of the seeds were dissected on ice and immediately frozen in a −80 °C refrigerator until RNA isolation. Each sample consisted of three biological replicates, and each replicate contained ten embryos.

Finally, a total of 24 samples from the two genotypes at four germination stages with three replicates were used for RNA sequencing. RNA was extracted using TRIzol reagent (Invitrogen, Life Technologies) following the standard protocol. The quality and quantity of the RNA were checked for integrity on an Agilent Technologies 2100 Bioanalyzer (Agilent Technologies, Santa Clara, CA, USA) according to their RNA Integrity Number (RIN) value. RNA-sequencing library construction was performed following the TruSeq RNA Sample Prep v2 protocol. The libraries were sequenced on an Illumina HiSeq 4000 platform with 150 bp paired-end reads at the Huazhong Agricultural University (Wuhan, China). The data analysis was conducted according to standard procedures; briefly, the raw data from the high-throughput sequencing were first filtered using cutadapt (v1.13) [45], and then clean reads were aligned to the maize reference genome (B73 AGPv4) using the HISAT2 (v2.1.0) tool [46]. Transcripts per kilobase of exon model per million mapped reads (TPM) values were calculated to represent the gene expression level. Differential expression analysis was performed between pairwise samples using the DESeq (2022) R package. The raw *p*-values were adjusted for multiple testing using the BH method [47], and genes were declared differentially expressed between two samples if the adjusted *p*-values were less than 0.05 and the fold-change was greater than 2. Finally, gene ontology (GO) and Kyoto Encyclopedia of Genes and Genomes (KEGG) enrichment analysis was performed using the packages ‘clusterProfiler’ [48] in R software [49], and the GO term was declared significantly enriched if the *q*-value were less than 0.05.

### 2.5. WGCNA Co-Expression Network Construction Module Identification

Weighted gene co-expression network analysis was performed using the WGCNA package (Version 4.0.2) in R software [50]. This analysis is based on the TPM value of all 24 samples from the two genotypes during germination and filtering out the DEGs with a low expression (TPM < 1 in all samples). The basic process of WGCNA analysis was as follows: First, the soft thresholding power was determined using the R function pickSoftThreshold when the fitted curve first approached 0.9. Second, gene modules were detected using a hierarchical clustering tree according to the calculated soft threshold. Third, correlation analysis between modules and samples, between modules and activities of CAT and SOD, was calculated to explore specific modules. Finally, hub genes in target modules were extracted according to the criteria of |KME| > 0.9 (module eigengene-based connectivity) and |GS| > 0.2 (gene significance). Function annotations and GO annotations of the target hub genes were obtained from the NCBI website (https://www.ncbi.nlm.nih.gov/, accessed on 15 August 2023) and the Gramene website (https://maize-pangenome.gramene.org/, accessed on 15 August 2023).

### 2.6. QTL Analysis and Investigate Effects of the Target Genes

The DH population was genotyped using genotyping by sequencing (GBS) technology, and an ultra-high-density linkage map of the DH population was developed as previously described [44]. In brief, the ultra-high-density linkage maps contained 64,553 high-quality SNP and INDEL markers and were integrated into 1101 bins based on the linkage and crossover among these markers. QTL analysis was performed using the composite interval mapping method (CIM) in Windows QTL Cartographer v2.5 [51]. The average germination rate at 15 days after sowing under low temperatures was used for QTL analysis, and the LOD threshold was set to 3.0. The confidence interval (CI) of QTLs was defined as the 2-LOD interval flanking the QTL peak. Additionally, the effects of the target genes were investigated based on flanking markers of the gene locus, and a single-tailed student *t*-test was performed to check the difference. Significant marker-trait associations were declared at *p* < 0.05.

## 3. Results

### 3.1. Germination Rate Analysis of the Two Genotypes

Germination rates of Q1 and Ye478 at low and optimum temperatures were used to study low-temperature tolerance during germination. At optimum temperature conditions, the germination rate of both the Ye478 and Q1 could reach 100%, but the Ye478 was earlier to reach (Figure 1A,C). In the low-temperature condition, the germination rate of Ye478 reached 100% on the fifth day after sowing, while Q1 reached only 60% by the 14th day after sowing and no longer changed (Figure 1B,D). These results indicated that the Ye478 could be more tolerant to low-temperature stress during germination compared with the Q1.

### 3.2. RNA-Sequencing Analysis for the Two Genotypes during Seed Germination

To explore the molecular basis of LTGA, transcriptome analysis was conducted for Ye478 and Q1 at multiple low-temperature germination stages. In total, 24 libraries from the two genotypes at four germination stages with three biological replicates were constructed and analyzed. Based on RNA-sequencing analysis, 686.7 million clean paired-end (PE) reads were obtained, and the average number of PE reads per library was 28.6 million (Appendix A). Genes with normalized reads lower than 0.5 TPM were removed from this analysis. Finally, 21632, 21699, 22173, and 23351 transcripts were found to be expressed at the 0d, 1d, 3d, and 5d stages of Ye478, respectively. In a similar vein, 21776, 21935, 22568, and 23506 transcripts were also identified in the samples from the respective stages of Q1. Approximately 35.4% of expressed genes were in the 0.5–5 TPM range, and 56.5% of expressed genes were in the range 5–100 TPM (Figure 2A).

Principal component analysis for the 24 samples was performed based on expression quantity, and the results are depicted in Figure 2B. The results showed that the biological replicates of each sample could be clearly clustered together, suggesting the reliability of RNA-sequencing data. Significant differences between Ye478 and Q1 were observed at each of the 0d, 1d, 3d, and 5d stages, and the variation tendency of the two genotypes was consistent through the germination stages, suggesting that the overall transcriptome profiling is determined by both genotype and germination stage.

### 3.3. Identification of DEGs at the Low-Temperature Germination Stage for Ye478

To identify genes with altered expression levels under low-temperature stress during germination for Ye478, we conducted pairwise comparisons between the control stage (0d) and the three low-temperature stress stages (1d, 3d, and 5d), respectively. In total, 6466 DEGs (4068 up- and 2398 down-regulated), 10519 DEGs (6227 up- and 4292 down-regulated), and 12527 DEGs (7532 up- and 4995 down-regulated) were identified at the 1d, 3d, and 5d stages compared with the 0d stage, respectively (Figure 3A,B). In order to further investigate the molecular mechanism of Ye748 responding to low-temperature stress, we combined these DEGs into one gene set and then conducted GO enrichment analysis. A total of 7964 up-regulated genes in this gene set were enriched in 98 biological process (BP) terms, 28 molecular function (MF) terms, and 50 biological process (CC) terms (*q*-value < 0.05), and the top 30 terms are shown in Figure 4A. These top terms are mainly involved in ribosome synthesis (“ribosome”, “structural constituent of ribosome”, “cytosolic ribosome”, “ribosomal subunit”, “large ribosomal subunit”, “cytosolic large ribosomal subunit”, “cytosolic small ribosomal subunit”, “small ribosomal subunit”), and hydrogen peroxide metabolism and function (“hydrolase activity”, “response to oxidative stress”, “hydrogen peroxide metabolic process”, “hydrogen peroxide catabolic process”, “reactive oxygen species metabolic process”, “hydrolase activity”, “acting on peroxide as acceptor”, “peroxidase activity”). In addition, 5010 down-regulated DEGs in this gene set were enriched in 39 BP terms, 14 MF terms, and 20 CC terms, most of which were related to the mRNA metabolism (Figure 4B). These results implied that ribosome synthesis and hydrogen peroxide metabolism were enhanced and mRNA metabolism was weakened at the low-temperature germination stage for Ye478.

### 3.4. Identification of DEGs at the Low-Temperature Germination Stage for the Q1

Similarly, we studied the DEGs between the 0d stage and the three low-temperature stress stages for Q1 and identified 2061 DEGs (1594 up- and 467 down-regulated), 5775 DEGs (3866 up- and 1909 down-regulated), and 6673 DEGs (4487 up- and 2186 down-regulated) at the 1d, 3d, and 5d stages compared with the 0d stage, respectively (Figure 3A,C). These DEGs were also combined into one gene set, which contained 6060 up-regulated genes and 2653 down-regulated genes. GO enrichment analysis showed that the up-regulated genes were enriched in 150 BP terms, 34 MF terms, and 28 CC terms, and the top 30 terms are shown in Figure 4C. Based on these results, the most significant terms involved in hydrogen peroxide metabolism and response to abiotic stimulus (“response to oxidative stress”, “hydrogen peroxide metabolic process”, “hydrogen peroxide catabolic process”, “reactive oxygen species metabolic process”, “acting on peroxide as acceptor”, “peroxidase activity”, “antioxidant activity”, “monooxygenase activity”, “response to oxygen-containing compounds”, “response to temperature stimulus”, “response to temperature stimulus”). Additionally, GO enrichment analysis was also performed for the down-regulated DEGs and significantly enriched three BP terms (mRNA metabolic process, mRNA processing, regulation of cell morphogenesis), three MF terms (cysteine dioxygenase activity, helicase activity, chromatin binding), and one CC term (plasma membrane protein complex) (Figure 4D). These results implied that hydrogen peroxide metabolism and response to abiotic stimulus were enhanced, and mRNA metabolism was weakened at the low-temperature germination stage for Q1.

### 3.5. Identification of DEGs between the Two Genotypes

Pairwise comparisons were performed to identify the DEGs at the same low-temperature germination stage between the two genotypes, and the numbers of DEGs are shown in Figure 3A. At the 0d stage, 4338 and 5340 genes were found to be significantly up- or down-regulated, respectively, in samples of the Ye478 compared with Q1. Of these, the up-regulated DEGs were significantly enriched in 22 BP terms and two MF terms (Figure 5A). In these BP terms, “response to heat”, “response to temperature stimulus”, “response to abiotic stimulus” and “cellular response to heat” were directly associated with temperature response, and “response to hydrogen peroxide”, “response to oxygen-containing compounds”, “response to abscisic acid”, and “response to alcohol” could be associated with tolerance. These results suggested differences in biological function related to temperature responses between the two genotypes. In addition, GO enrichment analysis was also conducted for down-regulated DEGs at the T0 stage. Based on the results, 111 BP terms, 49 MF terms, and 14 CC terms were significantly enriched, and the top 30 terms are shown in Figure 5B. These top terms are mainly involved in MF in terms of hydrolase and oxidoreductase activity and BP in terms of amino acid and glycometabolism, which could be involved in stress response.

Similarly, at the 1d stage, 3168 and 3976 genes were significantly up- or down-regulated in Ye478, respectively. At the 3d stage, the up- or down-regulated genes in Ye478 were 2727 and 3934, respectively. At the 5d stage, the up- or down-regulated genes in Ye478 were 2643 and 2396, respectively (Figure 3A). We combined DEGs from these three stages into one gene set, which included 4655 up-regulated DEGs and 5722 down-regulated DEGs. This gene set was considered to be the DEGs between the two genotypes under low-temperature stress during germination, and GO enrichment analysis was carried out. Based on the results, the up-regulated DEGs were significantly enriched in ten BP terms and three CC terms (Figure 5C). Most of these BP terms were associated with ribosome biogenesis and processing, and the three CC terms were “nucleolus”, “preribosome” and “nuclear lumen”, which suggested that the Ye478 has more ribosome synthesis under low-temperature stress during germination. For those down-regulated DEGs, a total of 106 BP terms, 58 MF terms, and nine CC terms were significantly enriched, and the top 30 terms are depicted in Figure 5D. Based on the results, most of the top BP and MF terms were related to response to oxidative stress and secondary metabolic processes, for example “response to oxidative stress”, “hydrogen peroxide metabolic process”, “reactive oxygen species metabolic process”, “oxidoreductase activity”, “peroxidase activity”, “secondary metabolic process”, “lipid biosynthetic process” and “response to external stimulus”. These results indicated that the Ye478 conducted more ribosome functions under low-temperature stress, while the Q1 conducted more stress responses.

### 3.6. WGCNA Analysis

In order to study the specific gene modules and genes that are highly associated with LTGA, we performed a WGCNA using the expression values (TPM) based on all 24 samples. After filtering out the DEGs with low expression (TPM < 1 in all samples), a total of 19753 genes were retained for the WGCNA analysis. The soft thresholding powers with β = 10 (R^2^ = 0.91) were selected for further cluster dendrogram analysis (Appendix A). Finally, these genes were divided into 16 modules (labeled with different colors), as shown in Figure 6A. Subsequently, the correlation analysis of these modules with samples was conducted (Figure 6B). Based on the results, we found that the MEbrown and MEred modules were highly positively correlated with the Ye478 at the low-temperature germination stage. And the MEgreen module was highly positively correlated with the Q1 across low-temperature stress, while the MEblack module was highly negatively correlated with the Q1. Similarly, some modules were significantly correlated with specific samples, for example, between the MEpink module with the sample Q0d (r = 0.78), between the MEyellow module with the sample Q0d (r = 0.79), between the MEtan module with the sample Y0d (r = −0.71), between the MEturquoise module with the sample Y0d (r = 0.95), between the MEsalmon module with the sample Q1d (r = 0.72), between the MEpurple module with the sample Q3d (r = 0.80), and between the MEblue module with the sample Q5d (r = 0.77). In addition, we also analyzed the relationship between these modules and oxidoreductase activity and found that the MEbrown module was significantly correlated with CAT and SOD activity (Figure 6B). All of these specific modules could be associated with LTGA.

Notably, the MEbrown module was highly correlated with the Ye478 at the low-temperature germination stage and was significantly correlated with CAT and SOD activity, which could be the key gene module for LTGA. GO enrichment analysis showed that a total of 2398 genes in the MEbrown module were significantly (*q* < 0.05) enriched in 26 BP terms, 11 MF terms, and six CC terms (Table 1). Of them, the BP terms were mainly involved in the regulation of cell cycle and phosphorylation, and the MF terms were mainly associated with trans-membrane transporters and cyclin-dependent protein kinases. KEGG enrichment analysis found that nine pathways were significantly enriched in the MEbrown module and mainly involved in the biosynthesis of plant secondary metabolites (Appendix A). In addition, the MEred module was highly related to the Ye478 across the three low-temperature germination stages, which could be important to LGTA. GO and KEGG enrichment analyses found that 1090 genes were significantly enriched in 61 GO terms (33 BP terms, eight MF terms, and 20 CC terms) and two pathways, and almost all of them were related to ribosome synthesis (Table 1 and Appendix A). Meanwhile, the MEgreen and MEblack modules had a continually negative and positive correlation with Ye478 at the four germination stages, which indicates that the genes in these two modules were continually down- and up-regulated in Ye478. GO and KEGG enrichment analyses found that the genes in the MEgreen module are mainly enriched in GO terms of ions trans-membrane transport and pathways of oxidative phosphorylation, carbon metabolism, and amino acid metabolism. For the MEblack module, we did not identify significantly enriched GO term and KEGG pathway.

Meanwhile, we analyzed hub genes in the MEbrown, MEred, MEgreen, and MEblack modules based on the principle that |KME| > 0.9 and |GS| > 0.3 and identified 414, 188, 239, and 137 hub genes, respectively. These four gene modules and hub genes could be essential to low-temperature tolerance during seed germination.

### 3.7. QTL Analysis and Investigate Effects of the Target Genes

QTL analysis was performed for the germination rate on the 15th day after sowing at low temperatures using the DH population. We identified a major QTL with LOD scores of 7.2, which explanted 20.6% of phenotypic variation and spanned a physical region from 191.3 to 209.5 Mb on chromosome 1 (Table 2). And this QTL could increase germination rate under low-temperature stress when the allele is derived from the Ye478 (Table 2). In this major QTL interval, we, respectively, found three, four, three, and zero hub genes from MEbrown, MEred, MEgreen, and MEred modules (Table 3), of which six genes (*Zm00001d031951*, *Zm00001d031953*, *Zm00001d031651*, *Zm00001d031667*, *Zm00001d031617*, and *Zm00001d031992*) have significant expression differences between the Ye478 and Q1 (Appendix A). In addition, correlation analysis between flanking marker and low-temperature germination rate found significant differences at eight gene locus (*Zm00001d031951*, *Zm00001d031953*, *Zm00001d031560*, *Zm00001d031640*, *Zm00001d031651*, *Zm00001d031667*, *Zm00001d031617*, and *Zm00001d031992*), especially at the *Zm00001d031951* and *Zm00001d031953* locus (Figure 7). Of them, *Zm00001d031951* and *Zm00001d031953* are involved in the glutathione metabolic process; *Zm00001d031651* is involved in the mitochondrial intermembrane space; *Zm00001d031617* is involved in oxidoreductase activity; and *Zm00001d031667* and *Zm00001d031992* do not have function annotations (Table 3). All of these hub genes could be the candidate genes for LTGA in this study.

## 4. Discussion

Low-temperature germination tolerance is necessary for spring sowing maize, especially in high-altitude and high-latitude regions [52]. However, conventional genetic breeding of maize mainly focuses on increasing yield potential and usually results in the loss of specific alleles. Understanding molecular mechanisms and discovering excellent alleles will be helpful for the directional breeding improvement of cultivated varieties. In this study, Q1 is an important inbred line for hybrid corn production in Southwest and Northwest China, but it has poor germination ability at low-temperature stress, while another elite inbred line, Ye478 retains the characteristic of low-temperature germination tolerance (Figure 1). Therefore, exploring low-temperature tolerance alleles in Ye478 and transferring these alleles into Q1 is an effective strategy to create low-temperature tolerance germplasm resources and directionally improve the LTGA of maize varieties.

Low-temperature tolerance is a complex biological process that usually involves sophisticated regulatory mechanisms. Of these, ribosome synthesis and processing have been found to correlate with low-temperature tolerance in many plants, including *Arabidopsis* [53,54], maize [38,39,40], and rice [55,56]. In our results, at the low-temperature germination stage, a large number of ribosome synthesis and processing biological processes were found only in Ye478 but not in Q1 (Figure 4A). More importantly, based on the WGCNA and correlation analysis, we also found that one of the most significant modules, Mered, mainly functioned ribosome synthesis and processing (Table 1). These results suggested the importance of normal ribosome synthesis for adapting to low-temperature stress. In addition, numerous genes involved in the cell cycle were considered to be closely associated with low-temperature responses in many plants, such as maize [37] and rice [36,57]. In this study, we identified a significant module, MEbrown, using WGCNA analysis, which was highly correlated with Ye478 at the low-temperature germination stage and CAT and SOD activity. And GO analysis for the MEbrown module found most biological processes related to cell cycle (Table 1), which indicated that the normal cell cycle function of Ye478 could be essential to withstand low-temperature stress, and this function could be influenced by CAT and SOD activity. Except for these potential molecular mechanisms, ROS-scavenging enzyme systems, including SOD and CAT, were also important mechanisms for low-temperature tolerance [34]. Here, we identified numerous biological processes related to ROS metabolism in both Ye478 and Q1 when they were exposed to low temperature during germination, which suggests ROS could be a response molecule to low-temperature stress.

Exploiting low-temperature tolerance genes and understanding the molecular basis of LTGA will be helpful for directional breeding improvement. In maize, many genes have been demonstrated to confer low-temperature tolerance. For example, two IF genes (*ZmDREB1/CB*, *bZIP68*) and two MAPK genes (*ZmRR1*, *ZmMPK8*) regulate low-temperature tolerance only at the seedling stage, and another IF gene *ZmICE1* regulate low-temperature tolerance both in the seedling and germination stages [5,6,7,8]. However, in DEGs between Ye478 and Q1, we did not identify significant enriched GO terms related to IF and MAPK. In contrast, we found ribosome synthesis, ROS-scavenging enzyme systems, and the cell cycle could be the main mechanisms for low-temperature tolerance in this study. Meanwhile, WGCNA and correlation analysis also found many hub genes related to ribosome synthesis and cell cycle (Table 1), and these hub genes have a higher expression level in the Ye478 at the low-temperature germination stage, which indicated the importance of these hub genes to withstand low-temperature stress. Additionally, we identified a major QTL associated with LTGA that spanned a physical region from 191.3 to 209.5 Mb on chromosome 1 based on a DH population. In this major QTL interval, we found three and four hub genes from MEbrown and MEred modules, respectively (Table 3). Of these, *Zm00001d031951* and *Zm00001d031953* are involved in the glutathione metabolic process, and glutathione is a well-known chemical for scavenging ROS [34,35], which further indicates the importance of these genes to low-temperature stress. The *Zm00001d031617* is involved in oxidoreductase activity, which could be related to the ROS-scavenging system, which was an important mechanism for low-temperature tolerance [32,33]. Therefore, these genes could be the key candidate genes to withstand low temperatures in this study.

## 5. Conclusions

LTGA is essential to the earlier sowing of spring maize and high-altitude maize. To investigate the molecular mechanism and explore candidate genes for this trait, we performed transcriptome analysis for inbred lines Ye478 and Q1 at low-temperature germination stages and constructed a co-expression network with WGCNA. The results showed that 7964 up- and 5010 down-regulated DEGs of Ye478 were identified after low-temperature treatment, and 6060 up- and 2653 down-regulated DEGs of Q1 were identified. GO enrichment analysis found ribosome synthesis and hydrogen peroxide metabolism of Ye478 were enhanced under low-temperature stress, while hydrogen peroxide metabolism of Q1 was enhanced. DEGs pairwise comparisons between the two genotypes found Ye478 performed more ribosome functions at low-temperature stress compared with Q1. WGCNA analysis MEbrown and MEred modules could be target modules related to LTGA, and they are mainly involved in ribosome synthesis and cell cycle, respectively. In QTL analysis based on the DH population constructed by Ye478 and Q1, we identified a major QTL explaining 20.7% of the phenotype variance. In the QTL interval, ten hub genes in the MEbrown, MEred, MEgreen, and MEblack modules were significantly associated with low-temperature germination rate, of which two hub genes related to glutathione metabolism and one hub gene related to oxidoreductase activity could be the candidate genes for LTGA. These biological functions and key genes will be helpful in understanding the regulatory mechanism and directional improvement variety for LTGA.

## Figures and Tables

**Figure 1 genes-14-01917-f001:**
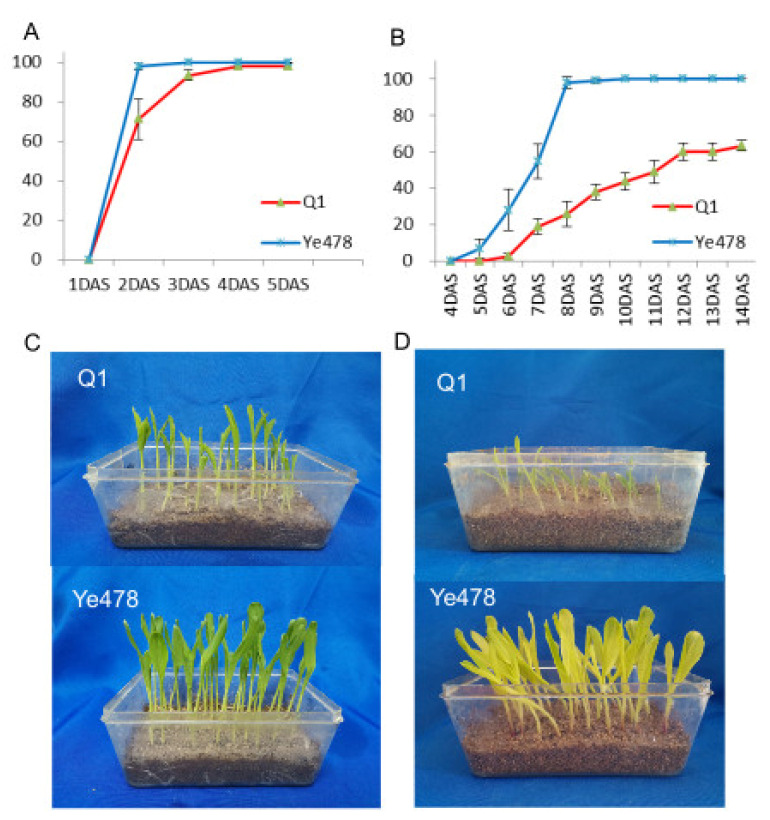
Germination rates of Ye478 and Q1 under optimum and low temperatures during germination. (**A**) germination rates under optimum temperature. (**B**) germination rates under low temperature. (**C**,**D**) the final germination performance under optimum and low temperatures, respectively.

**Figure 2 genes-14-01917-f002:**
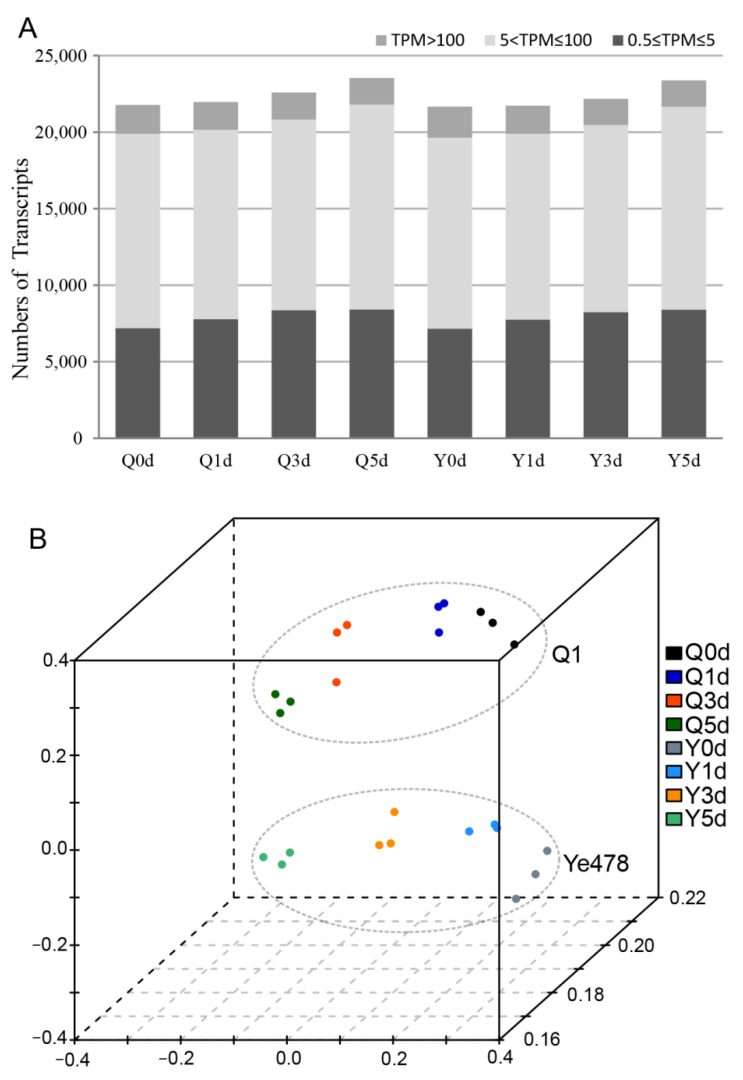
Global gene expression profiling of the two genotypes in all germination stages. (**A**) Numbers of detected transcripts in each sample. (**B**) Principal component analysis of the RNA-sequencing data. And Q0d, Q1d, Q3d, and Q5d represent germination stages of the Q1 at 0d, 1d, 3d, and 5d stages, respectively; Y0d, Y1d, Y3d, and Y5d represent germination stages of the Ye478 at 0d, 1d, 3d, and 5d stages, respectively.

**Figure 3 genes-14-01917-f003:**
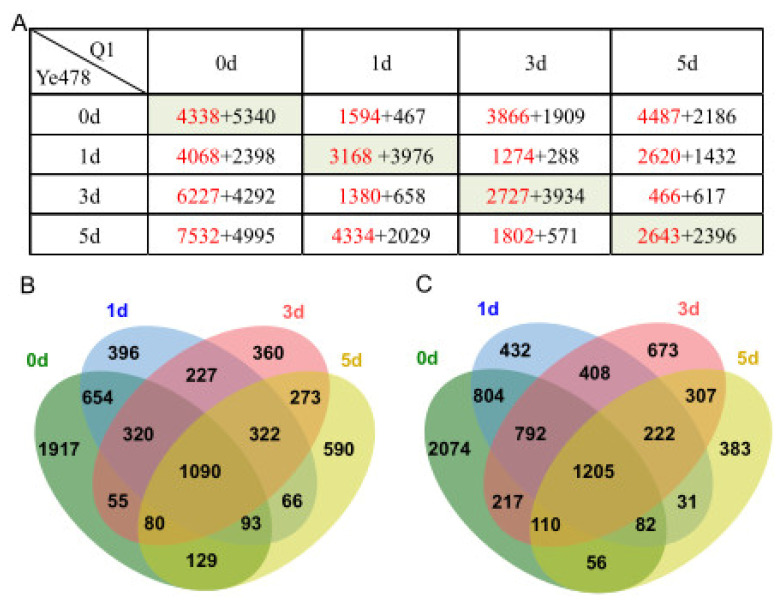
Numbers of DEGs of the two genotypes at each germination stage. (**A**) Numbers of DEGs between the two genotypes at the same germination stage (on the diagonal line), among different stages for Q1 (above the diagonal line), and among different stages for Ye478 (below the diagonal line), and the red and black numbers are the up- and down-regulated DEGs when the sample in the later stage is compared with the sample in the earlier stage. (**B**,**C**) the overlapping numbers of the up-and-down-regulated DEGs between Ye478 and Q1 at the same germination stages, respectively.

**Figure 4 genes-14-01917-f004:**
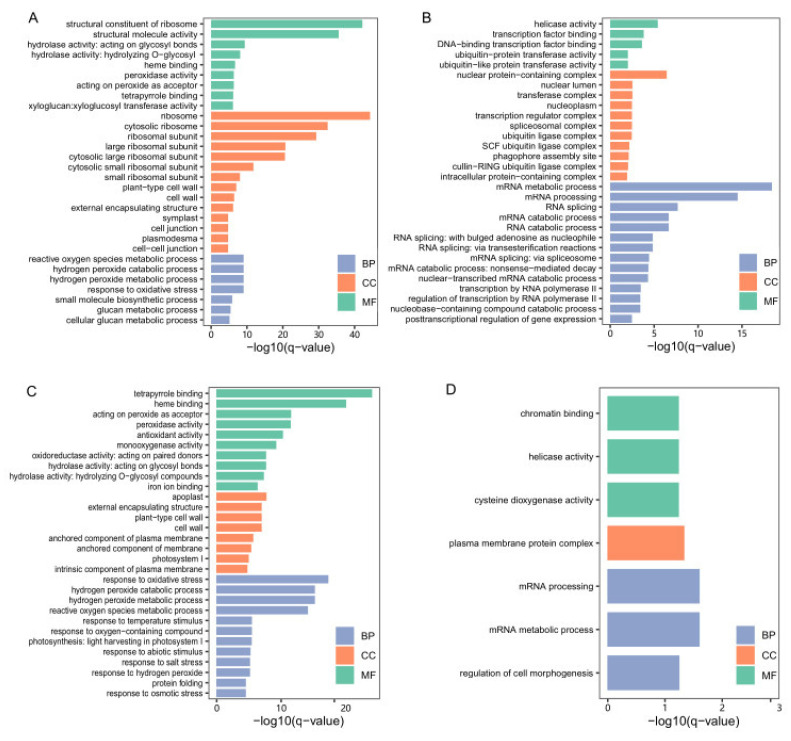
GO enrichment analysis of DEGs under low-temperature stress compared with the control stage. (**A**) The top 30 GO terms of up-regulated DEGs for Ye478. (**B**) The top 30 GO terms of down-regulated DEGs for Ye478. (**C**) The top 30 GO terms of up-regulated DEGs for Q1. (**D**) The significant enriched GO terms of down-regulated DEGs for Q1.

**Figure 5 genes-14-01917-f005:**
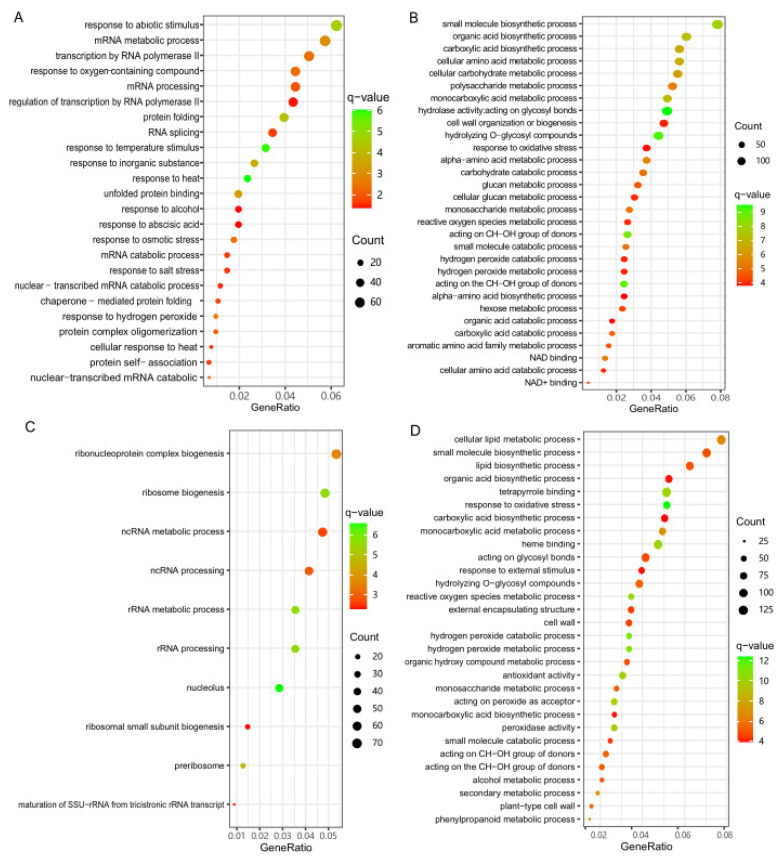
GO enrichment analysis of DEGs between the two genotypes at the same germination stage. (**A**) The significant enriched GO terms of up-regulated DEGs at T0 stage. (**B**) The top 30 GO terms of down-regulated DEGs at the T0 stage. (**C**) The significant enriched GO terms of up-regulated DEGs at the low-temperature stages. (**D**) The top 30 GO terms of down-regulated DEGs at low-temperature stages.

**Figure 6 genes-14-01917-f006:**
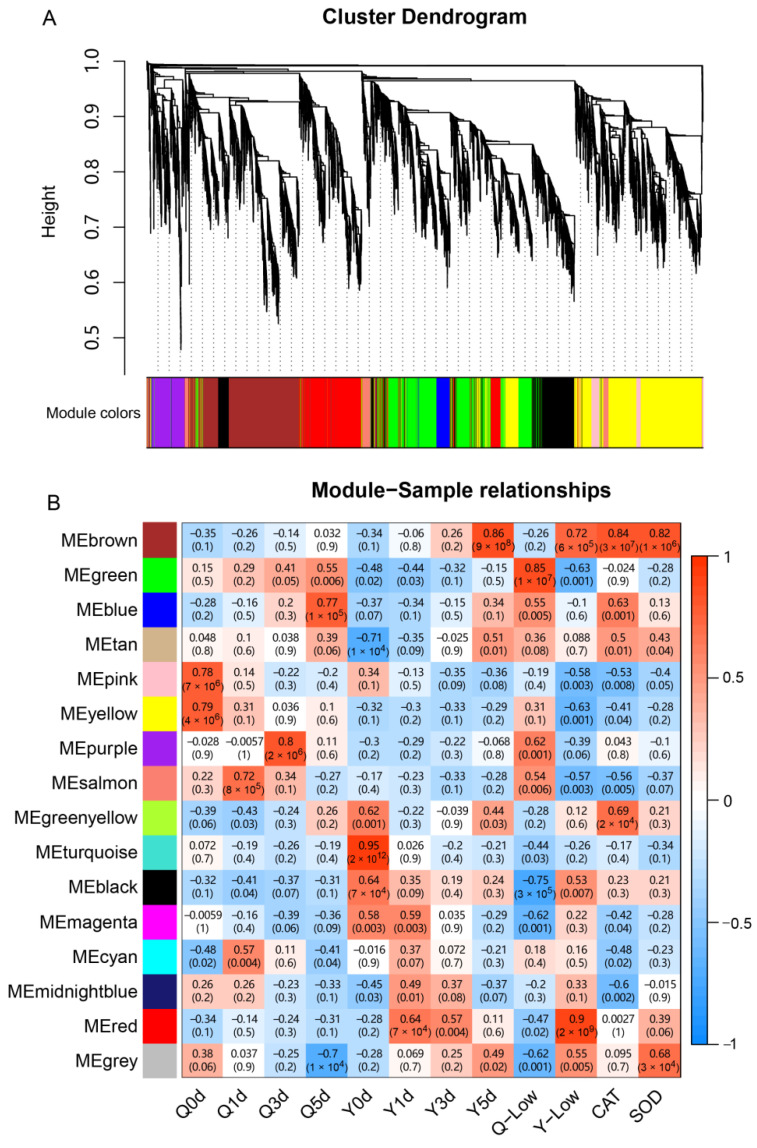
WGCNA of genes in Ye478 and Q1 at each germination stage. (**A**) Hierarchical dendrogram reveals co-expression modules identified by WGCNA. The major tree branches constitute 16 modules, labeled with different colors. (**B**) The correlation coefficient and correlation significance between each module with sample and oxidoreductase activity.

**Figure 7 genes-14-01917-f007:**
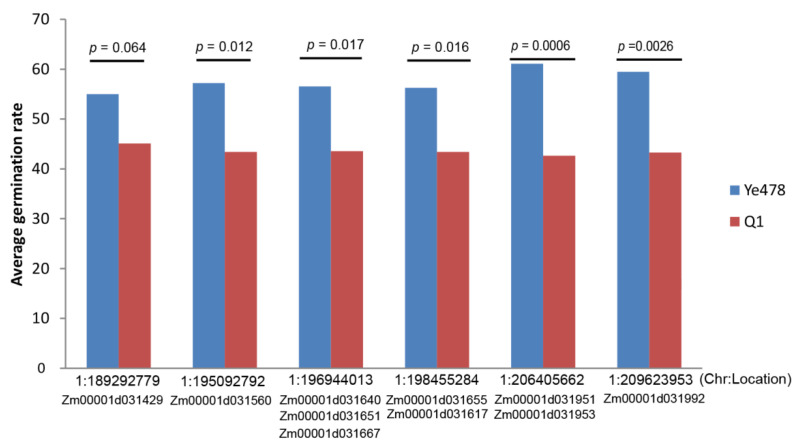
Low-temperature germination effects of candidate genes in the DH population.

**Table 1 genes-14-01917-t001:** Top GO terms of the target modules.

Ontology	ID	Description	*q*-Value
MEbrown:			
BP	GO:0043549	regulation of kinase activity	1.5 × 10^5^
BP	GO:0045859	regulation of protein kinase activity	1.5 × 10^5^
BP	GO:0000079	regulation of cyclin-dependent protein serine/threonine kinase activity	1.5 × 10^5^
BP	GO:1904029	regulation of cyclin-dependent protein kinase activity	1.5 × 10^5^
BP	GO:0001932	regulation of protein phosphorylation	4.1 × 10^5^
BP	GO:0042325	regulation of phosphorylation	4.1 × 10^5^
BP	GO:0071900	regulation of protein serine/threonine kinase activity	4.1 × 10^5^
MF	GO:0046983	protein dimerization activity	9.2 × 10^5^
BP	GO:0007049	cell cycle	1.8 × 10^4^
BP	GO:0051338	regulation of transferase activity	2.3 × 10^4^
BP	GO:0051726	regulation of cell cycle	2.3 × 10^4^
BP	GO:0040008	regulation of growth	1.0 × 10^3^
BP	GO:0000278	mitotic cell cycle	1.2 × 10^3^
BP	GO:0048638	regulation of developmental growth	1.3 × 10^3^
BP	GO:0044770	cell cycle phase transition	1.3 × 10^3^
MEred:			
CC	GO:0005840	ribosome	2.8 × 10^140^
MF	GO:0003735	structural constituent of ribosome	8.4 × 10^137^
MF	GO:0005198	structural molecule activity	4.8 × 10^124^
CC	GO:0022626	cytosolic ribosome	9.8 × 10^89^
CC	GO:0044391	ribosomal subunit	2.5 × 10^85^
CC	GO:1990904	ribonucleoprotein complex	3.2 × 10^72^
CC	GO:0022625	cytosolic large ribosomal subunit	6.5 × 10^56^
CC	GO:0015934	large ribosomal subunit	6.1 × 10^54^
CC	GO:0022627	cytosolic small ribosomal subunit	1.3 × 10^32^
CC	GO:0015935	small ribosomal subunit	7.0 × 10^30^
BP	GO:0042254	ribosome biogenesis	1.5 × 10^25^
BP	GO:0022613	ribonucleoprotein complex biogenesis	3.0 × 10^23^
BP	GO:0042273	ribosomal large subunit biogenesis	1.5 × 10^17^
BP	GO:0042255	ribosome assembly	7.8 × 10^17^
BP	GO:0002181	cytoplasmic translation	6.7 × 10^16^
MEgreen:			
CC	GO:0016469	proton-transporting two-sector ATPase complex	8.6 × 10^8^
CC	GO:0033178	proton-transporting two-sector ATPase complex	7.4 × 10^7^
CC	GO:0033176	proton-transporting V-type ATPase complex	9.7 × 10^7^
CC	GO:0033180	proton-transporting V-type ATPase, V1 domain	1.3 × 10^5^
CC	GO:0098796	membrane protein complex	1.6 × 10^3^
CC	GO:1990204	oxidoreductase complex	4.4 × 10^2^
MF	GO:0044769	ATPase activity, coupled to transmembrane movement of ions	1.7 × 10^7^
MF	GO:0046961	proton-transporting ATPase activity, rotational mechanism	1.7 × 10^7^
MF	GO:0019829	ATPase-coupled cation transmembrane transporter activity	1.3 × 10^4^
MF	GO:0042625	ATPase-coupled ion transmembrane transporter activity	1.3 × 10^4^
MF	GO:0015078	proton transmembrane transporter activity	8.3 × 10^4^
MF	GO:0016627	oxidoreductase activity	3.9 × 10^2^

**Table 2 genes-14-01917-t002:** QTLs for germination rate at 15th days after sowing.

Chr.	Peak (cM)	LOD	Add	R^2^ (%)	2 LOD (cM)	Location (Mb)
1	105.0	5.0	−23.7	13.8	93.2–106.1	118.5–174.8
1	128.8	7.2	19.0	20.6	117.8–139.7	191.3–209.5
2	114.2	3.4	9.4	8.9	109.3–119.1	223.6–229.7

**Table 3 genes-14-01917-t003:** The hub genes of target modules in the major QTL interval.

GeneID	KME	GS	NR Annotation	GO Term	Location (Chr. 1)
MEbrown:					
Zm00001d031655	0.90	0.41	uncharacterized protein LOC100281072	regulation of transcription	197,545,631–197,549,052
Zm00001d031951	0.98	0.63	uncharacterized protein LOC100272476	glutathione metabolic process	207,697,908–207,854,723
Zm00001d031953	0.92	0.69	uncharacterized protein LOC100272476	glutathione metabolic process	207,932,227–207,935,640
MEred:					
Zm00001d031429	0.96	0.84	pfkB-like carbohydrate kinase family protein	structural constituent of ribosome	189,856,343–189,862,099
Zm00001d031560	0.95	0.81	60S ribosomal protein L32	structural constituent of ribosome	194,266,307–194,270,465
Zm00001d031640	0.90	0.66	hypothetical protein SETIT_2G267400v2	*	197,108,702–197,111,394
Zm00001d031651	0.92	0.88	Mitochondrial import inner membrane translocase subunit Tim8	mitochondrial intermembrane space	197,393,450–197,395,554
MEgreen:					
Zm00001d031667	0.92	−0.42	hypothetical protein Zm00014a_031611	*	196,224,574–196,228,047
Zm00001d031617	0.92	−0.50	Discolored-paralog2	oxidoreductase activity	198,119,074–198,130,785
Zm00001d031992	0.93	−0.49	indeterminate domain p1	*	208,942,917–208,950,782

“*” represent that the genes have no GO annotation in the Gramene website.

## Data Availability

Not applicable.

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
