# Peer review of "Transcriptome Analysis and QTL Mapping Identify Candidate Genes and Regulatory Mechanisms Related to Low-Temperature Germination Ability in Maize"

_genes, 2023, doi:10.3390/genes14101917_

Round 1
Reviewer 1 Report
This work provides a new insight into modern maize breeding, I have some questions below, and the authors may consider in the revised manuscript.
For the detection of DEGs, can you use a stricter threshold? For some comparisons, the authors detected more than 10,000 DEGs, which means 50% of expressed genes are DEGs. It’s too many.
The authors compared transcriptome profiling from control and three low-temperature stress stages. How can you exclude the bias caused by plant growth? If compare the data from different time points at optimum temperature conditions, I am wondering if you still can detect thousands of DEGs. Comparing data from optimum and low temperatures seems more interesting.
Have you checked the expression pattern over the four-time points? The expression of some genes may continue to increase or decrease, are they low-temperature response genes?
Are there any module genes that enrich the DEGs or are involved in any pathways?
The authors identified three QTLs for germination rate and focused on the major one. They found several hub genes located at this major QTL. What is the expression fold change of these hub genes? Do they have the most significant expression difference? Have you checked the other gene expression within this locus?
In Figure 7, please provide a statistical method.
Author Response
Response to Reviewer 1 Comments:
Dear Reviewer,
Thank you for your comments concerning our manuscript entitled “Transcriptome analysis and QTL mapping identify candidate genes and regulatory mechanisms related to low-temperature germination ability in maize” (genes-2629974). Those comments are all valuable and very helpful for revising and improving our paper, as well as the important guiding significance to our researches. We have studied comments carefully and have made correction which we hope meet with approval. Revised portion are marked using the track changes functionality of our Word in the paper, and our responses to comments are marked in red as shown below. The main corrections in the paper and the responds to the reviewer’s comments are as flowing:
Comment#1: For the detection of DEGs, can you use a stricter threshold? For some comparisons, the authors detected more than 10,000 DEGs, which means 50% of expressed genes are DEGs. It’s too many.
Response: Thank you very much for reviewer’s comment. Your suggestion is reasonable, and the number of detected DEGs will decrease based on stricter threshold. We have carefully evaluated this treatment of DEGs detection, and not hope to re-calculate DEGs based on a stricter threshold and the reasons were in below. The stricter threshold could decrease number of DEGs artificially, but it could lose some key genes that were related target trait. And adjusting the detection threshold has little effect for the conclusion of this study. Therefore, we do not suggest re-calculating DEGs based on a stricter threshold.
Comment#2: The authors compared transcriptome profiling from control and three low-temperature stress stages. How can you exclude the bias caused by plant growth? If compare the data from different time points at optimum temperature conditions, I am wondering if you still can detect thousands of DEGs. Comparing data from optimum and low temperatures seems more interesting.
Response: Thanks for your valuable comment. That is a very good and reasonable suggestion. However, at the same time point, the developmental stages under low-temperature and optimum temperature are not consistent; for example, the seedlings have emerged under optimum temperature at the third days after sowing, but not under low temperature. Therefore, we do not suggest conducting this comparison between low and optimum temperatures at the same time point. Additionally, based on the reviewer’s comment, we found some inappropriate descriptions on DEGs under low-temperature stress for Q1/Ye478, and have revised it as “DEGs at low-temperature germination stage for the Q1/Ye478” in the revised manuscript.
Comment#3: Have you checked the expression pattern over the four-time points? The expression of some genes may continue to increase or decrease, are they low-temperature response genes?
Response: Thanks for your valuable comment. We agreed with reviewer's suggestion of checking the expression pattern of genes and detecting continually down- and up-regulated genes. In our study, we identified expression pattern of genes using WGCNA method, and divided these genes into 16 gene modules based on expression patterns. In addition, considering the reviewer’s valuable suggestion, we have added the analysis about MEgreen and MEblack modules which were continually down- and up-regulated in Ye478 according to correlation analysis between sample and modules.
Comment#4: Are there any module genes that enrich the DEGs or are involved in any pathways?
Response: Thanks for your valuable comment. We have added the KEGG analysis for four target gene modules (MEbrown, MEred, MEgreen and MEblack modules) in the revised manuscript.
Comment#5: The authors identified three QTLs for germination rate and focused on the major one. They found several hub genes located at this major QTL. What is the expression fold change of these hub genes? Do they have the most significant expression difference? Have you checked the other gene expression within this locus?
Response: Thanks for your valuable comment. We have showed the expression level of these hub genes in the Table S2. Moreover, considering the reviewer’s valuable comment, we have checked the expression fold change of these hub genes, and found that four hub genes (Zm00001d031655, Zm00001d031429, Zm00001d031560 and Zm00001d031640) were not differentially expressed genes between the two genotypes. We have revised the description about these genes in the revised manuscript.
Comment#6: In Figure 7, please provide a statistical method.
Response: Thanks for your comment. We used the single-tailed student T-test method to check the difference in our study, and we have added the statistical method in Materials and Methods in the revised manuscript.

Reviewer 2 Report
Dear authors,
this study is very interesting with scientific importance. The manuscript is about the transcriptome analysis and QTL mapping identify candidate genes and regulatory mechanisms related to low-temperature germination ability (LTGA) in maize. The authors found three and four hub genes in MEbrown and MEred modules, of which two hub genes (Zm00001d031951, Zm00001d031953) related to glutathione metabolism and one hub gene (Zm00001d031560) related to ribosome synthesis could be the candidate genes for LTGA.
In the manuscript, introduction and objectives are well and clear written. The materials and methods are given in details. The results obtained and presented in 7 figures and 3 tables are relevant to the proposed objectives. The discussion is appropriate in the context of the results. The conclusions are supported by the results. The references are mostly recent and representative in the field of study.
Before accepting of the manuscript, following parts have to be corrected:
Author names and affiliations have to be in accordance with the Instructions for Authors (Genes_template file).
page 2, third paragraph:
desaturated fatty acid (DFC) content > desaturated fatty acid content (DFC)
in the entire manuscript where you have two consecutive references:
[20-21] > [20,21]
[5, 7] > [5,7]
Subsection titles in the entire manuscript have to be corrected in accordance with the Instructions for Authors.
Figure 6B > numbers are not readable, increase font size of correlation coefficients and correlation significances
Table 1, title: Mebrown and Mered > MEbrown and MEred
in Table 1 and Table 3: Mered: > MEred:
Table 3 has to be moved from Discussion section to Results section.
Discussion, second paragraph:
Arabidopsis > Arabidopsis
Author Contributions have to be written in accordance with the Instructions for Authors.
Minor editing of English language required.
Author Response
Response to Reviewer 2 Comments:
Dear Reviewer,
Thank you for your comments concerning our manuscript entitled “Transcriptome analysis and QTL mapping identify candidate genes and regulatory mechanisms related to low-temperature germination ability in maize” (genes-2629974). Those comments are all valuable and very helpful for revising and improving our paper, as well as the important guiding significance to our researches. We have studied comments carefully and have made correction which we hope meet with approval. Revised portion are marked using the track changes functionality of our Word in the paper, and our responses to comments are marked in red as shown below. The main corrections in the paper and the responds to the reviewer’s comments are as flowing:
Comments #1: Author names and affiliations have to be in accordance with the Instructions for Authors (Genes_template file).
Response: We thank the reviewer for his comment. We have added the detail institution name of authors in the revised manuscript, and will contact the assistant editor to correct the information in template file. In addition, the authors of Lei Du and Yongzhong Liu worked in both Hubei Hongshan Laboratory and Huazhong Agricultural University, and we written their institution names according to contributions of institution.
Comments #2: page 2, third paragraph: desaturated fatty acid (DFC) content > desaturated fatty acid content (DFC)
Response: Thanks for your careful review. We are very sorry for our negligence, and have carefully revised this mistake.
Comments #3: in the entire manuscript where you have two consecutive references: [20-21] > [20,21] ; [5, 7] > [5,7]
Response: Thanks for your careful review. We are very sorry for our negligence, and checked the reference citation in detail in our manuscript. We have revised these mistakes in the revised manuscript.
Comments #4: Subsection titles in the entire manuscript have to be corrected in accordance with the Instructions for Authors.
Response: Thanks for your comment. We guess that the reviewer is asking us to improve the detailed institution information of authors, and we have added the detail institution information of authors in the revised manuscript. If not, please point out in detail, thanks very much.
Comments #5: Figure 6B > numbers are not readable, increase font size of correlation coefficients and correlation significances
Response: Thanks for your comment. We have replaced the Figure 6 with a clearer figure and increase font size.
Comments #6: Table 1, title: Mebrown and Mered > MEbrown and MEred; in Table 1 and Table 3: Mered: > MEred:
Response: Thanks for your careful review. We are very sorry for our negligence, and have carefully revised these mistakes in the revised manuscript.
Comments #7: Table 3 has to be moved from Discussion section to Results section.
Response: Thanks for your comment. We have moved the Table 3 to results section in the revised manuscript.
Comments #8: Discussion, second paragraph: Arabidopsis > Arabidopsis
Response: Thanks for your careful review. We are very sorry for our negligence, and have carefully revised this mistake in the revised manuscript.
Comments #9: Author Contributions have to be written in accordance with the Instructions for Authors.
Response: Thanks for your careful review. We have corrected the Author Contributions in the revised manuscript according to the template file, and have contacted the assistant editor to correct some information according to contributions.

Round 2
Reviewer 1 Report
I appreciate the authors answered most of my questions.
I don't have any other questions.